# Effect of the Adhesive System on the Properties of Fiberboard Panels Bonded with Hydrolysis Lignin and Phenol-Formaldehyde Resin

**DOI:** 10.3390/polym14091768

**Published:** 2022-04-27

**Authors:** Viktor Savov, Ivo Valchev, Petar Antov, Ivaylo Yordanov, Zlatomir Popski

**Affiliations:** 1Faculty of Forest Industry, University of Forestry, 1797 Sofia, Bulgaria; 2Faculty of Chemical Technologies, University of Chemical Technology and Metallurgy, 1757 Sofia, Bulgaria; ivoval@uctm.edu (I.V.); yordanov@uctm.edu (I.Y.); 3Welde Bulgaria, 5600 Troyan, Bulgaria; popski@welde.bg

**Keywords:** wood-based panels, fiberboards, adhesive system, hydrolysis lignin, phenol–formaldehyde resin, optimization, hot-pressing

## Abstract

This study aimed to propose an alternative technological solution for manufacturing fiberboard panels using a modified hot-pressing regime and hydrolysis lignin as the main binder. The main novelty of the research is the optimized adhesive system composed of unmodified hydrolysis lignin and reduced phenol–formaldehyde (PF) resin content. The fiberboard panels were fabricated in the laboratory with a very low PF resin content, varying from 1% to 3.6%, and hydrolysis lignin addition levels varying from 7% to 10.8% (based on the dry wood fibers). A specific two-stage hot-pressing regime, including initial low pressure of 1.2 MPa and subsequent high pressure of 4 MPa, was applied. The effect of binder content and PF resin content in the adhesive system on the main properties of fiberboards (water absorption, thickness swelling, bending strength, modulus of elasticity, and internal bond strength) was investigated, and appropriate optimization was performed to define the optimal content of PF resin and hydrolysis lignin for complying with European standards. It was concluded that the proposed technology is suitable for manufacturing fiberboard panels fulfilling the strictest EN standard. Markedly, it was shown that for the production of this type of panels, the minimum total content of binders should be 10.6%, and the PF resin content should be at least 14% of the adhesive system.

## 1. Introduction

The growing demand for eco-friendly wood-based panels with lower environmental footprints and reduced hazardous emissions of volatile organic compounds, such as free formaldehyde from the finished wood composites, have imposed new stricter regulations and requirements on both researchers and industrial practice, related to the development of sustainable, “green” composites [1,2,3,4,5]. The production of fiberboards, with an estimated global production of more than 104 million m^3^ in 2020, is the second largest worldwide, surpassed only by the production of plywood [6]. In the production of dry-process fiberboards, which accounts for about 74% of the total output, the problem with the hazardous formaldehyde emissions from the finished panels is also relevant [7,8]. A viable approach to solving this issue is using sustainable, bio-based, formaldehyde-free wood adhesives to partially or completely replace the conventional synthetic formaldehyde-based wood adhesives, such as urea–formaldehyde (UF), melamine–urea formaldehyde (MUF), and phenol–formaldehyde (PF) resins, commonly used in the panel industry [9,10,11,12,13,14]. Successful attempts for the development of bio-based adhesives, including modified condensed and hydrolyzed tannins, proteins, starch, lignin, carbohydrates, etc., have been reported [15,16,17,18,19,20]. The main drawbacks of using 100% bio-based adhesive formulations for bonding wood composites are related to the need for additional modification of natural raw materials to improve their chemical reactivity, the deteriorated dimensional stability and mechanical properties of the wood-based panels produced, and the need to modify the technological parameters, e.g., through the extension of pressing time. In terms of industrial utilization, significant positive results have been obtained in producing wood-based panels, mainly particleboards, with tannin-based bio-adhesives [21,22,23].

Lignin is an amorphous, three-dimensional complex biopolymer, composed of phenylpropanoid units linked by intramolecular bonds, and the second most abundant natural material, surpassed only by cellulose. Lignin contains a large number of functional groups, e.g., aliphatics, phenolic, hydroxyl, and carbonyl groups, and acts as a natural binder in wood, being the main component of the middle lamella connecting wood cells [24]. In the production of wet process fiberboards, the properties of the panels are mainly due to the lignin bonds arising from the hot-pressing. This makes lignin a particularly promising bio-based adhesive for manufacturing dry-process fiberboards.

Significant quantities of lignin by-products, estimated to approximately 100 million tons per year, are generated worldwide, mostly as a waste and side streams of the pulp and paper industries, of which only about 2% is used for conversion into value-added products [25,26,27]. The enhanced valorization and commercial utilization of that lignin will support the transition to circular economy [28,29]. Lignin can be extracted from lignocellulosic biomass by applying physical, chemical, and biological treatment methods. Depending on the method by which they are obtained, the residual lignin products, i.e., technical lignins, are sulfate (Kraft) lignin, sulfite lignin (lignosulfonate), organosolv lignin and hydrolysis lignin. Although Kraft lignin is the most widespread globally, much of it is burned in the factories where it is obtained, which regenerates some of the chemical reagents used and produces heat and energy [30,31]. The main drawbacks for using lignosulfonates in wood adhesive formulations are related to the higher number of impurities, e.g., high sulfur and ash content, compared to the Kraft lignin, and the deteriorated hydrophobic properties of the wood-based panels, so it is recommended to be used in combination with synthetic binders with or without additional cross-linking [32,33,34,35,36,37,38,39,40,41,42]. Although there are some studies on the use of organosolv lignin in wood adhesives, mostly as a partial replacement of phenol in PF resins, its wider use is limited due to the significantly smaller quantities [43], compared to Kraft lignin and lignosulfonates. In the case of hydrolysis lignin (HL), these shortcomings are greatly avoided [44]. As HL is a by-product of bioethanol production, its amount is expected to increase worldwide [45,46].

Previous efforts to recover lignin have been focused mainly on its modification, its use as a substitute for phenol in lignin–phenol–formaldehyde (LPF) resins, used primarily in the production of plywood or the use of lignin for biofuels [47,48,49,50,51,52,53]. There are also successful attempts to produce fiberboards with lignin as a binder. Previous studies on the use of HL in wood adhesives showed that if it is introduced in the dry state in the pulp, lignin cannot be retained and deteriorated the properties of the panels [44]. It was determined that when using a traditional hot-pressing cycle with first high and subsequent decreasing pressure, the addition of lignin as a substitute of formaldehyde-based adhesives leads to a deterioration of the panel properties [54]. Good results were achieved by using HL with a very small content of PF resin as an auxiliary binder. PF resin was mainly used to improve the retention of lignin in the pulp, and to enhance the binding reactions of between lignin and wood fibers [55]. The cited study did not fully clarify the effect of the total binder content and the effect of the ratio of lignin to PF resin in the adhesive system.

This work aimed to investigate the effect of the total binder content and the optimal ratio between hydrolysis lignin and PF resin in the production of fiberboards using a modified hot-pressing regime. The main novelty is a derivation of the optimal composition of the adhesive system with maximum HL content when using a modified hot-pressing regime. The modification of the hot-pressing regime aims for maximal utilization of the adhesive abilities of the lignin, in contrast to the classic hot-pressing cycles, which are developed for formaldehyde synthetic resins.

## 2. Materials and Methods

In this study, PF resin was used as an auxiliary binder to retain the technical HL until the condensation and activation process occurred. The PF resin was chosen because of its better lignin convergence and higher temperature resistance than UF and MUF resins [56,57,58,59], Figure 1.

The present study used a classical approach for mixing PF resin and lignin. Still, a modified hot-pressing cycle was applied to make optimal use of the adhesion capabilities of lignin.

Industrially produced pulp obtained by the thermomechanical refining method in the factory Kronospan-Bulgaria EOOD (Veliko Tarnovo, Bulgaria) was used. The pulp was composed of 40% hardwood (beech and oak) and 60% softwood (pine). The pulp was characterized by a bulk density of 29 kg·m^−3^, fiber lengths varying from 1120 to 1280 μm, and a moisture content of 11.2% (factory data). The PF resin used was manufactured by Prefere Resins Romania SRL (Rasnov, Romania) and provided by Welde Bulgaria PLC (Troyan, Bulgaria). The PF resin had the following characteristics: dry solids content 46%, viscosity—358 MPa·s; brix 72.7 and acid factor (pH)—6.8.

The technical HL was produced from high temperature diluted sulfuric acid hydrolysis of sawdust and softwood and hardwood chips to sugars. The chemical composition of lignin was determined by the standard TAPPI methods [60,61]. The C, N, S and H analysis was performed by using Elemental Analyzer Euro EA 3000 (EuroVector, Pavia, Italy). After fractionation, only HL from the fraction below 100 μm was used.

In interior design and furniture production, thin and ultra-thin wood-based composites are increasingly used in a variety of end uses [62,63]. Therefore, the target thickness of the laboratory panels was set to 4 mm. Twelve fiberboard panels with dimensions 200 mm × 200 mm × 4 mm were manufactured in laboratory conditions, divided into two series with different adhesive systems. The target density of the laboratory-made fiberboard panels was 850 kg·m^−3^.

Due to the use of a bio-based binder, i.e., HL, the total binder content was higher compared to the conventional adhesive systems, composed only from thermosetting formaldehyde-based resins [64]. Thus, the total binder content used in this work was 10% and 12%. The higher content of binders when an adhesive system with the participation of lignin is used leads to a significant deterioration in the appearance of the panels, the need to extend the press factor and a very slight improvement in the properties of fiberboards [65,66]. The amount of PF resin in the adhesive system also varied and was 10%, 20% and 30%. The manufacturing parameters of fiberboards bonded with HL and PF resin are given in Table 1.

The HL and PF resin were adjusted to a concentration of 30%. Then they were mixed and almost immediately sprayed into the pulp. A high-speed glue blender at 850 rpm (laboratory prototype, University of Forestry, Sofia, Bulgaria) with needle-shaped blades was used. The adhesive formulation was injected through a nozzle with a diameter of 1.5 mm at a pressure of 0.4 MPa. The whole gluing process had a duration of 60 s.

The hot-pressing was performed on a laboratory press “Servitec-Polystat 200 T” (Servitec Maschinenservice GmbH, Wustermark, Germany). The hot-pressing temperature applied was 200 °C. The pressing was carried out in two stages with subsequent cooling. The first stage was performed at a pressure of 1.2 MPa and lasted 360 s. The second stage was carried out at a pressure of 4.0 MPa for 120 s. Cooling was carried out while maintaining the high pressure (4.0 MPa) up to a temperature below 100 °C. The cooling time was 360 s. That hot-pressing regime was chosen due to a previous study aimed at optimizing the pressing time using HL and PF resin [55]. In that study, an increase in the press factor for the second stage above 30 s·mm^−1^ is unjustified. Preliminary studies indicate that when an adhesive system from HL and PF resin is used, a deterioration in the waterproof properties of the panels is observed at hot-pressing temperatures below 200°. That is confirmed by other similar studies [65,66]. The optimal parameters of the hot-pressing pressure have also been established experimentally. During the first stage, at a pressure above 1.0 MPa, it is difficult to separate the gas mixture from the panels. At a pressure below 4.0 MPa, it is difficult to compress the fibers to the final thickness and density of the panels.

The fiberboard panels were conditioned for 10 days at a room temperature of 20 ± 2 °C and a relative humidity of 65%.

The physical and mechanical properties of the laboratory-fabricated fiberboard panels (Figure 2) were determined according to the EN standards [67,68,69,70]. Eight test samples were used for each property. A precision laboratory balance Kern (Kern & Sohn GmbH, Balingen, Germany) with an accuracy of 0.01 g was used to measure the mass of the test specimen. Digital calipers with a 0.01 mm accuracy were used to determine the dimensions of the test samples. Thickness swelling (TS) and water absorption (WA) tests were performed by the weight method after 24 h of immersion in water. A universal testing machine, Zwick/Roell Z010 (ZwickRoell GmbH & Co. KG, Ulm, Germany), was used to determine the mechanical properties of the panels.

Regression analysis was used to analyze the effect of the total binder content and the participation of PF resin in the adhesive system on the properties of the composites, and the following regression model was derived (Equation (1)):(1)$$\hat{Y}=B_{0}+B_{1}X_{1}+B_{2}X_{2}+B_{12}X_{1}X_{2}$$
where $\hat{Y}$ is the predicted value of the given property;

*B*_0_, *B*_1_, *B*_2_, *B*_12_—regression coefficients;

*X*_1_, *X*_2_—the studied factors.

Stepwise regression with 1000 iterations was applied to perform optimization. For this purpose, specialized software “QstatLab”, version 6.0, was used.

## 3. Results and Discussion

The technical HL had the following chemical composition: 72.6% lignin, 25.5% cellulose, 2.8% ash content, 55.54 C, 0.74 S, 7.10 H, and 0.26 N. The ash content is relatively low, which leads to improved adhesion properties of technical HL. That is, the ash will not significantly impair the adhesive bonds. The presence of cellulose in the HL contributes to its properties as a binder [71,72].

The results obtained for the density of laboratory-fabricated fiberboard panels are presented in Table 2.

The density of the panels varied from 842 to 862 kg·m^−3^. The difference between the maximum and minimum density values was 2.4%, i.e., significantly below the permissible statistical error of 5%. This was also confirmed by the conducted ANOVA (Table 3). The test performed included the density data obtained for each test sample.

Therefore, it can be concluded that there was no statistically significant difference between the density of the individual fiberboards, thus it will not affect the other physical and mechanical properties.

The derived regression models, in explicit form, are presented in Table 3.

Table 4 shows that the obtained models are statistically significant. In all models, the calculated value of the Fisher’s criterion (F_cal_) is greater than the critical value (F_(0.05,3,2)_).

A graphical representation of the results obtained for the modulus of elasticity (MOE) of the fiberboard panels fabricated in this work is presented in Figure 3.

The overall improvement of the MOE values with the variation of the adhesive system used was 1.7 times. All manufactured panels bonded with an adhesive composition of HL and PF resin as an auxiliary binder met the standard requirements for dry-process fiberboards for general purpose and use in dry conditions—2700 N·mm^−2^ [70]. All panels, except for the fiberboards produced with 10% total binder content and 10% PF resin in the adhesive system, fulfilled the strictest requirements for this property, i.e., for fiberboards for load-bearing applications—3000 N·mm^−2^.

At a total binder content of 10% with an increase in the PF resin content from 10% to 30% in the adhesive system, an improvement in the MOE values of 1.4 times was observed. That improvement was most significant (by 1.3 times) when the PF resin content in the total adhesive system was increased from 10% to 20%. According to previous studies on the application of lignin as a binder in fiberboards, its incorporation could increase the stiffness of the panels, resulting in higher MOE values [19,71].

The subsequent increase in the content of PF resin from 20% to 30% in the adhesive system had a slight effect, i.e., MOE values improved by 1.2 times. Therefore, it can be concluded that at 10% total binder content, the content of PF resin in the adhesive system should be above 10%. Otherwise, the retention of HL in wood fiber mass is not complete, which results in decreased MOE values of the panels [54,55].

Fiberboard panels fabricated with 12% total binder content had higher MOE values than fiberboards bonded with 10% adhesives. Even at 10% content of PF resin in the adhesive system, a higher MOE was observed compared with the panels manufactured with 10% binder content, of which 30% was PF resin. With the increased PF resin content from 10% to 30% of the total adhesive system used, an improvement of MOE values by 1.09 times was observed. Similarly, more significant increase in MOE values by 1.05 times was determined when the percentage of PF resin content on the adhesive system was increased from 10% to 20%.

The optimal, maximum, MOE value of 4680 N·mm^−2^ was obtained at the upper limit values of the factors—12% total binder content and 30% PF resin content in the adhesive system (Figure 4).

The grey zone in Figure 4 represents the limitation, i.e., the standard requirement for MOE values of fiberboard panels used in load-bearing applications in humid conditions (3000 N·mm^−2^) [70]. Markedly, fiberboard panels, fulfilling this requirement, can be manufactured with a total binder content of at least 10.3% and a PF resin content in the adhesive system of at least 14%.

The determined MOE values of laboratory-fabricated fiberboard panels with a hydrolysis lignin content of more than 10% in the adhesive system are in accordance with the results obtained by Yotov et al. [44]. Comparable MOE values were also reported by other authors [54,71], who investigated the effect of lignin incorporation in adhesive systems, used for fiberboard manufacturing.

The results obtained for the bending strength (MOR) values of the laboratory-produced fiberboard panels bonded with HL and PF resin are presented in Figure 5.

The fiberboard panels produced in this work exhibited MOR values, varying from 27.38 to 52.65 N·mm^−2^. The improvement in MOR values of the panels with increasing total binder content was more significant compared to MOE, i.e., an overall improvement of 1.93 times was determined. All laboratory panels met the standard requirements for use in dry conditions—MOR value of at least 27 N·mm^−2^ [70]. All panels, except for the fiberboards manufactured with a total binder content of 10% and 10% PF resin in the adhesive system, also fulfilled the requirements for load-bearing applications and use in humid conditions—30 N·mm^−2^.

At the total binder content of 10%, the increased PF resin content in the adhesive system from 10% to 30%, resulted in improved MOR values by 1.36 times. The respective increase in MOR was 1.18 times when the PF resin content in the adhesive system was increased from 10% to 20%, and 1.15 times when it was increased to 30%, respectively. MOR values depend on individual fiber strength and bonding strength among wood fibers, i.e., better inter-fiber bonds will result in improved MOR of the panels. The positive effect of HL addition on the fiber surface on MOR values of fiberboards was also confirmed by other authors [19,66,73].

Fiberboard panels fabricated with 12% total binder content exhibited higher MOR values than the panels produced with 10% total binder content. The fiberboards with 12% binder content, of which 10% was PF resin, had 1.15 times higher MOR values than the panels with 10% total binder content, of which 30% was PF resin. Panels bonded with a 12% total binder content showed improved MOR values by 1.42 times when the PF resin content was increased from 10% to 30%.

The optimal, maximum value of the property of 52.65 N·mm^−2^ was obtained at the upper limit values of the factors—total binder content of 12% and 30% PF resin content in the adhesive system (Figure 6).

The grey zone in Figure 6 represents MOR values below 30 N·mm^−2^. In all other cases, the strictest standard requirement for this property for load-bearing applications and use in humid conditions was fulfilled [70]. To note, this requirement can be achieved with a minimum binder content of 10.22% and PF resin content in the adhesive system of at least 13.5%.

The results show that at the selected levels of variation of the total binder content and the content of PF resin in the adhesive system, the increase in the content of PF resin resulted in improved MOE and MOR values of the panels. However, it should be noted that in all experimental series, the HL is the main binder, and that is, a waste bio-based material is utilized. The improvement in the properties of fiberboards with increasing PF resin content should be due to better retention of lignin in the pulp and the interaction of lignin with PF resin. Which of the two causes dominates should be a subject of subsequent studies.

The results obtained for the MOR values of the panels, fabricated in this work, are in accordance with the findings of other authors [44], where an improvement by 1.7 times in this property was reported with an increase in the binder content from 6% to 10%. In another study [71], the increased lignin content in the adhesive system resulted in a significantly improved (almost twice) MOR values. The beneficial effect of HL addition on MOR values of fiberboards, produced by modifying the hot-pressing regime was also reported by Valchev et al. [55].

The results obtained for the IB strength of the fiberboard panels, bonded with adhesive system composed of HL and PF resin, is presented in Figure 7.

The internal bond (IB) refers to the bonding strength between individual fibers, which is of great importance because it ensures that the panels will not delaminate during post-processing. Internal bonding between wood fibers without synthetic resins is due to the hydrogen binding between fibers, condensation reaction of lignin [74,75,76], and crosslinking reactions between lignin and polysaccharides [77]. The formed covalent bonds between lignocellulosic polymers contribute to the formation of intermolecular forces which are stronger compared to the ones due to hydrogen bonds [78]. Furthermore, fibers with lignin-rich surfaces have a positive effect on the mechanical properties of the panels due to entanglement of the softened lignin caused by applied pressure and temperature, and supplemented by covalent bond formation [74,75,76,79].

The IB values of the fiberboard panels produced in this work varied from 0.66 to 1.32 N·mm^−2^. Overall, the IB values of fiberboards were improved with the addition of HL. The increased total binder content and PF resin content in the adhesive system resulted in an almost twice increase in IB strength. For the panels bonded with a 10% total binder content, the IB values increased from 0.66 to 0.81 N·mm^−2^ with the increased PF resin content in the adhesive system from 10% to 30%, i.e., an improvement of 1.23 times was determined. More significant improvement of the IB values was observed with the increased total binder content from 10% to 12%. The tendency of increased IB of the panels with increasing PF resin content in the adhesive system was also confirmed. However, the overall improvement of this property achieved by increasing the PF resin content in the adhesive system from 10% to 30% was only 9%.

All fiberboard panels bonded with HL as the main binder and PF resin as an auxiliary binder fulfilled the European standard requirements for general purpose fiberboards used in dry conditions—0.65 N·mm^−2^ [70]. With the exception of the panels manufactured with 10% total binder content, of which 10% was PF resin, all other panels met the strictest requirements for the property, i.e., for load-bearing applications and use in humid conditions—IB strength of at least 0.70 N·mm^−2^ [70].

The optimal, maximum value of the property of 1.32 N·mm^−2^ was obtained at the upper limit values of the factors—total binder content of 12% and PF resin content of 30% of the adhesive system (Figure 8). In order to meet the most stringent requirement of 0.70 N·mm^−2^, the total binder content should be at least 10.15%, and the PF resin content in the adhesive system should be at least 15%.

The improvement of the mechanical properties of the panels, MOE, MOR, and IB strength, with the increased total binder content and lignin content, was consistent with previous studies using lignin as a bio-based binder for fiberboard manufacturing [80,81,82,83].

However, the predominant effect of improving the properties of fiberboard panels is the increase in total binder content, which increases both the content of HL and PF resin. Although, the results show that the ratio of HL to PF resin is also significant. If the PF resin is the main binder, then in the used hot-pressing regime, the resin polymerization will occur at low pressure, i.e., before the final compression of the material has occurred. That will lead to fabricating panels with deteriorated properties.

Thickness swelling (TS) and water absorption (WA) are important physical properties of wood composites, strongly related to the dimensional stability of the panels, and providing an insight of panel behavior when used in humid conditions, especially in outdoor applications [84,85].

The results obtained for the TS (24 h) of the fiberboard panels bonded with HL and PF resin are presented in Figure 9.

As seen in Figure 8, the TS values of the laboratory-produced fiberboards varied from 34.33% to 20.40%. The increased total binder content resulted in reduced TS values by 1.68 times. All manufactured fiberboards fulfilled the standard requirement for panels used in dry conditions—TS value of 35% [70]. Except for the panels manufactured with a 10% binder content, of which 10% was PF resin, all other fiberboards met the standard requirement for use in humid environments, i.e., 30%. The main reasons for TS in fiberboards are the breakage of bonded areas among wood fibers and the recovery of compressed fibers [74]. Results from previous research works indicated that the addition of HL on the fiber surface resulted in improved dimensional stability of the panels, due to the hydrophobic nature of HL [19,54].

For the panels fabricated with 10% total binder content, the increased PF resin content in the adhesive system, from 10% to 30%, resulted in decreased TS values by 1.51 times. Much more significant improvement of TS was observed when the PF resin content in the adhesive system was increased from 10% to 20%, i.e., 1.36 times. The subsequent increase in the PF resin content resulted in an improvement of 1.11 times.

With regards to the panels, manufactured with 12% total binder content, the increased PF resin content in the adhesive system from 10% to 30% resulted in improved TS values of the panels by 1.23 times. Markedly, the improvement of the studied property was more significant when the PF resin content was increased from 10% to 20% in the adhesive system, i.e., 1.16 times.

It can be concluded that the increased binder content from 10% to 12%, and PF resin content in the adhesive composition from 20% to 30%, respectively, had a limited effect on the TS of the laboratory-made fiberboards bonded with HL and PF resin.

The optimal (minimum) value of the property was obtained again at the upper limit values of the factors (Figure 10). The grey zone in the figure represents TS values higher than 30%, i.e., values that do not meet the standard requirement for use in humid conditions [70]. To fulfill the stringent standard requirements, the panels should be fabricated with a total binder content of at least 10.6%, and the PF resin content in the adhesive system should be at least 14%.

The improvement of TS by increasing the total binder content is consistent with the results obtained in previous studies [44,54,55,66].

The results of the WA (24 h) of fiberboard panels, bonded with HL and PF resin, are presented in Figure 11.

Lignocellulosic materials absorb water by creating hydrogen bonds between water and hydroxyl groups of lignin, cellulose, and hemicellulose present in the cell wall [86,87]. The addition of lignin in the adhesive system reduces the WA of fiberboards due to the presence of aromatic rings and non-polar hydrocarbon chains in the lignin structure [88].

WA values of the fiberboard panels produced in this work varied from 73.27% to 54.05%. The variation of the total binder content and adhesive system composition resulted in reduced WA of the panels by 1.36 times. The reduced WA values with the incorporation of lignin in the adhesive formulation might be attributed to bulking the cell wall with lignin, which makes it hydrophobic [19,29].

For the panels manufactured with 10% total binder content, the most significant improvement of WA (1.21 times) was determined when the PF resin content in the adhesive system was increased from 10% to 20%. The subsequent increase in the PF resin content in the adhesive system from 20% to 30% had a more negligible effect, and the observed improvement was only 1.04 times.

The increased total binder content from 10% to 12% resulted in improved WA values of the laboratory panels. However, fiberboards fabricated with 10% total binder content, of which 20% was PF resin, exhibited WA values, comparable with the panels bonded with 12% total binder content, of which PF resin was 10%.

The optimal WA value was obtained at the upper limit values of the factors (Figure 12). As WA is not a standardized property of wood-based composites, the strictest restrictions were imposed on the other properties of the panels. Thus, it was found that fiberboards complying with the most stringent requirements, namely for load-bearing applications and use in humid conditions, can be manufactured with a minimum total binder content of 10.6%, where PF resin should be at least 14%. Therefore, the HL addition may constitute 86% of the adhesive system.

The results obtained for improving the waterproof properties of the fiberboard panels by increasing the lignin content are consistent with the findings reported in similar studies [65,80,81,82,83]. The results for the physical and mechanical properties of fiberboards manufactured with a modified adhesion system and hot-pressing cycle are comparable or better than the results reported in similar studies on the application of lignin as a bio-based wood adhesive [65,66,80,83]. Compared to the cited studies, an advantage of the technology used in this work is the absence of lignin modification, lower total lignin content, and lower density of the panels.

## 4. Conclusions

The HL is activated when modifying the hot-pressing regime, most likely by plasticization and subsequent condensation processes involving the PF resin. That overcomes the disadvantage of low lignin retention in the wood fiber mass. The proposed hot-pressing technology is easily feasible in industrial conditions with continuous presses. These presses have autonomous heating of the individual sections, so cooling will not lead to significant heat losses and additional costs. It should be emphasized that continuous presses with a cooling zone are already in operation. The fiberboard panels produced in this work from industrial wood fiber mass, bonded with an adhesive system composed of HL and PF as an auxiliary binder, exhibited excellent mechanical and hydrophobic properties, fulfilling the requirements of the relevant standard for application in humid conditions. Markedly, fiberboard panels meeting the most stringent standard requirements can be manufactured with a minimum total binder content of 10.6%, composed of at least 14% PF resin content, and at least 86% HL, respectively. It can be concluded that further increase in total binder content and PF resin content is not justified unless for manufacturing special purpose panels. The main disadvantage of the proposed technological solution is the long first stage of hot-pressing, which can be overcome by reducing the moisture content of the fiber mat. Future research should be focused on determining the optimal moisture content of the mat to enable lignin activation at reduced hot-pressing duration.

## Figures and Tables

**Figure 1 polymers-14-01768-f001:**
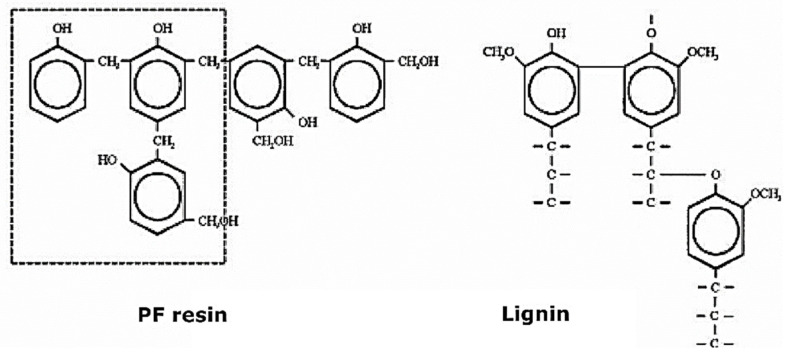
Structure of PF resin and lignin [59].

**Figure 2 polymers-14-01768-f002:**
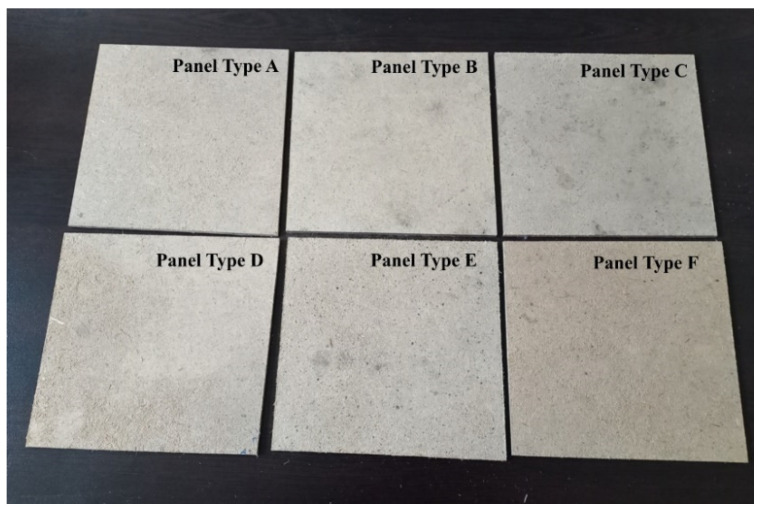
Fiberboard panels bonded with HL and PF resin: Type A—10% of binders, from which 10% was PF resin; Type B—10% of binders, from which 20% was PF resin; Type C—10% of binders, from which 30% was PF resin; Type D—12% of binders, from which 10% was PF resin; Type E—12% of binders, from which 20% was PF resin; Type F—12% of binders, from which 30% was PF resin.

**Figure 3 polymers-14-01768-f003:**
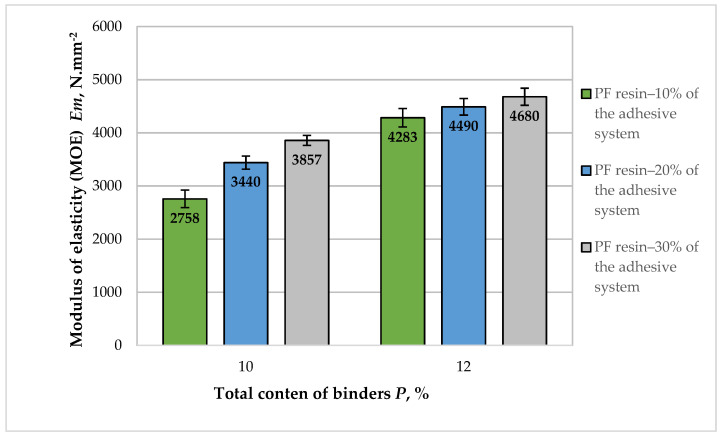
Modulus of elasticity (MOE) of fiberboard panels bonded with HL and PF resin.

**Figure 4 polymers-14-01768-f004:**
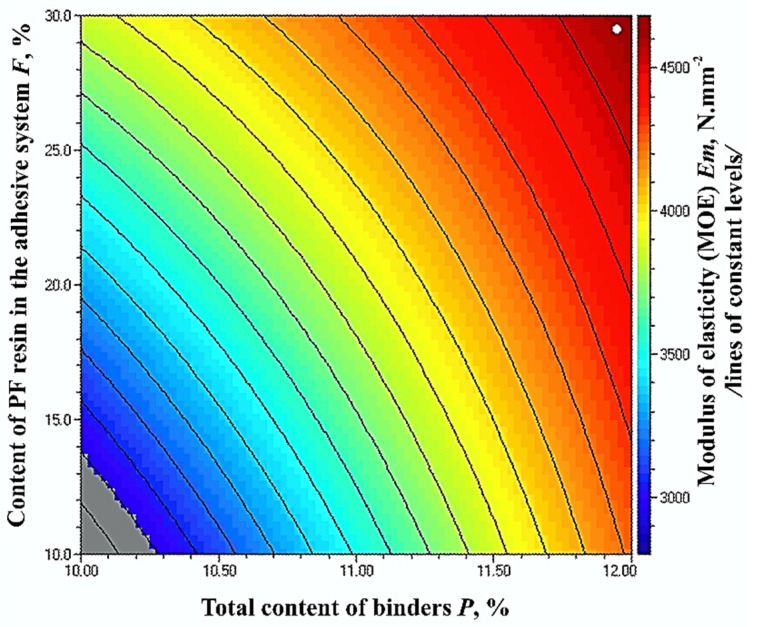
Effect of binder content and PF resin content in the adhesive system on the MOE values of fiberboard panels.

**Figure 5 polymers-14-01768-f005:**
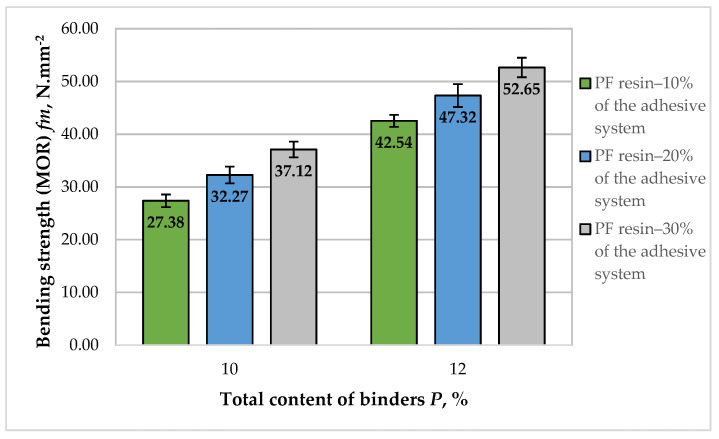
Bending strength (MOR) of fiberboard panels bonded with HL and PF resin.

**Figure 6 polymers-14-01768-f006:**
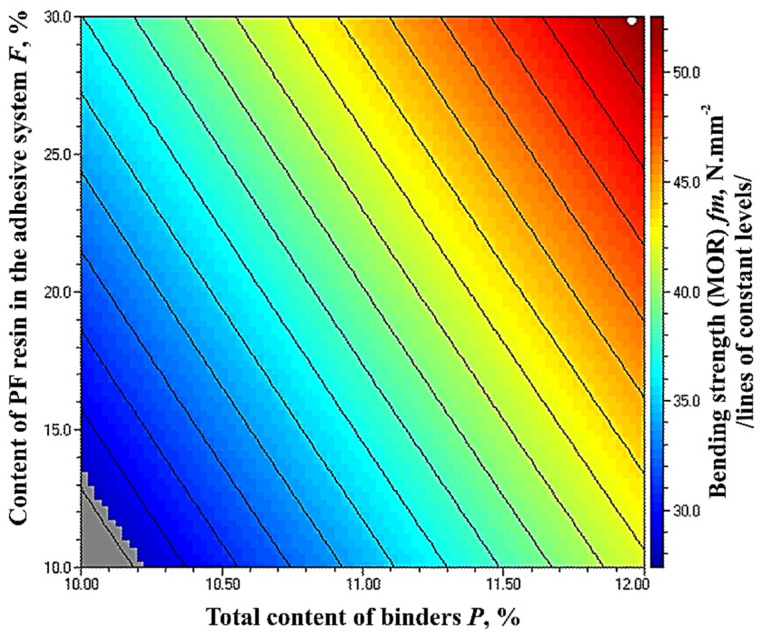
Effect of binder content and PF resin content in the adhesive system on the MOR values of fiberboard panels.

**Figure 7 polymers-14-01768-f007:**
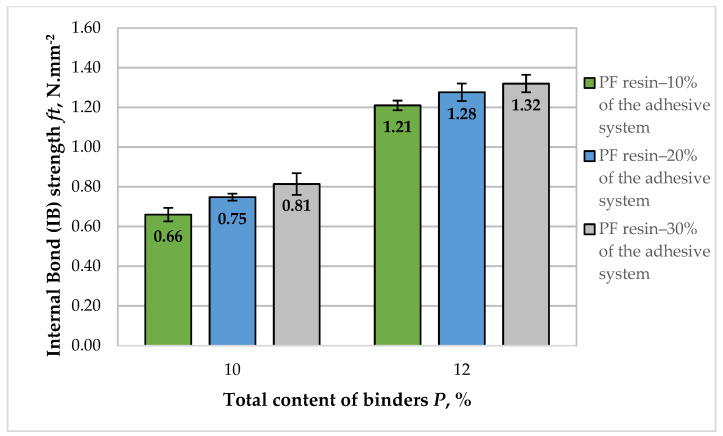
Internal bond (IB) of fiberboard panels bonded with HL and PF resin.

**Figure 8 polymers-14-01768-f008:**
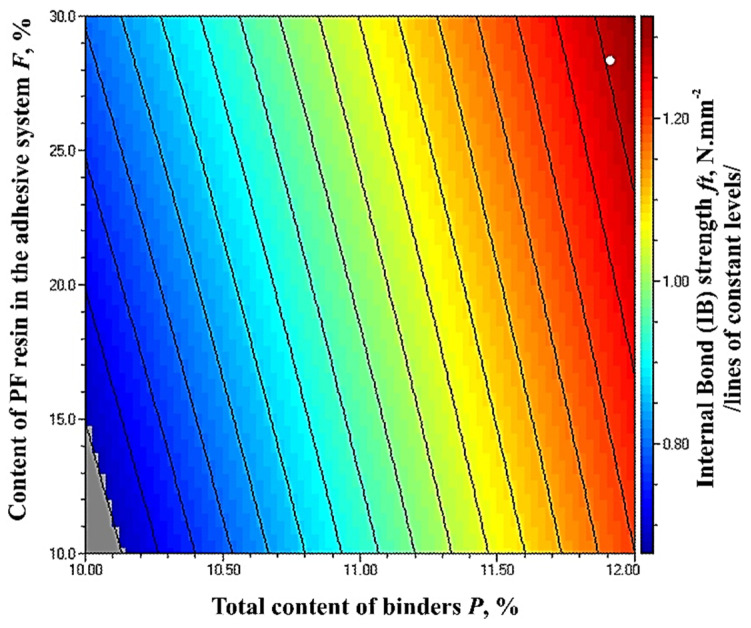
Effect of binder content and PF resin content in the adhesive system on the IB strength of fiberboard panels.

**Figure 9 polymers-14-01768-f009:**
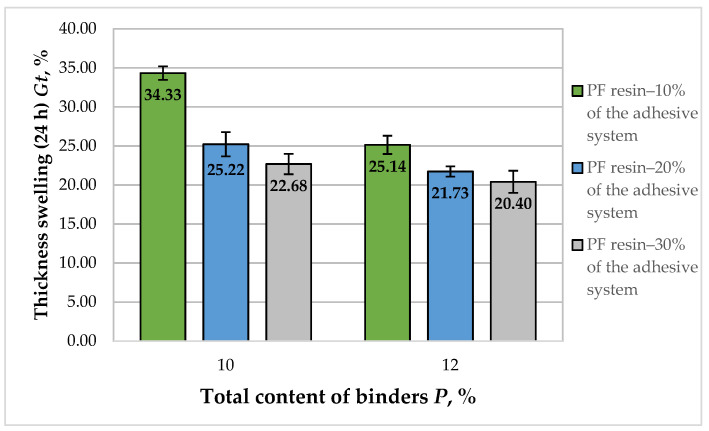
Thickness swelling (24 h) of fiberboard panels bonded with HL and PF resin.

**Figure 10 polymers-14-01768-f010:**
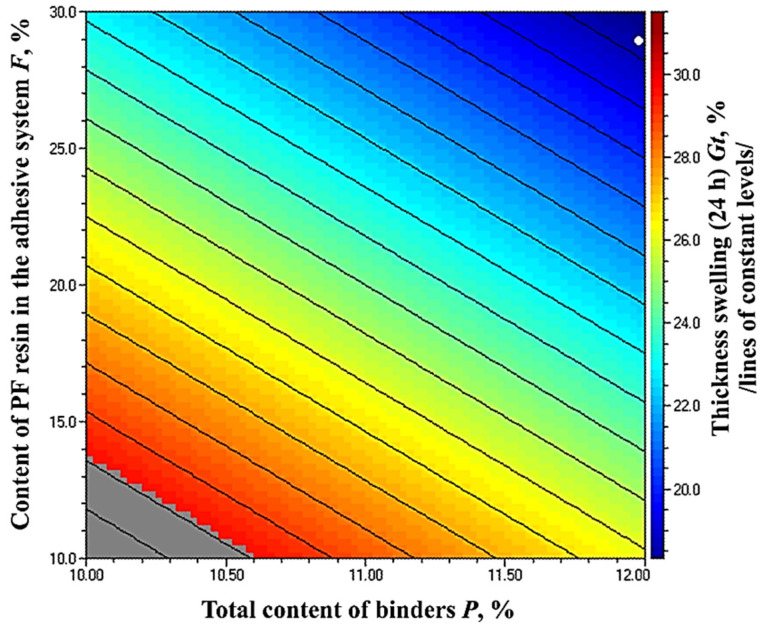
Effect of binder content and PF resin content in the adhesive system on the TS of fiberboard panels.

**Figure 11 polymers-14-01768-f011:**
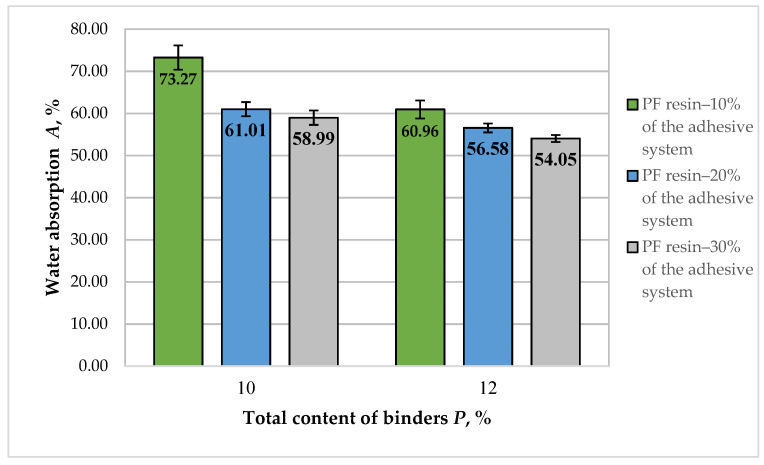
Water absorption (24 h) of fiberboard panels bonded with HL and PF resin.

**Figure 12 polymers-14-01768-f012:**
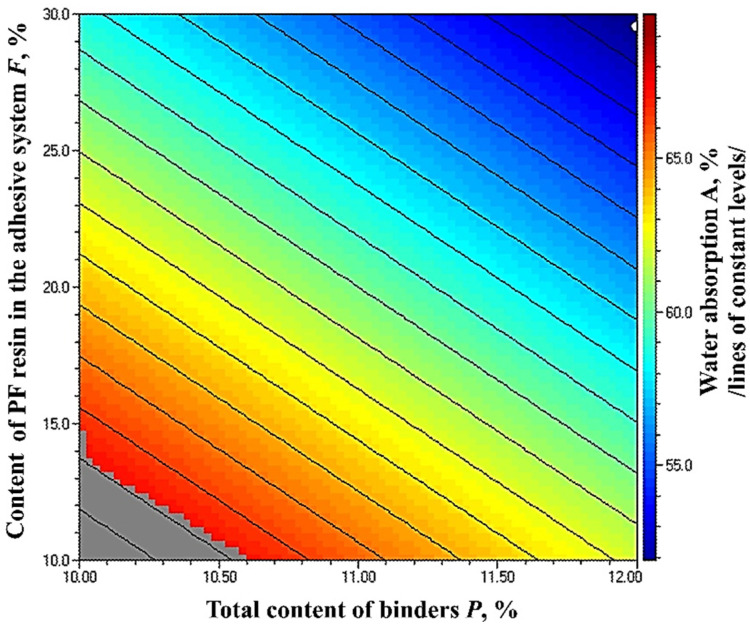
Effect of binder content and PF resin content in the adhesive system on WA (24 h) of fiberboard panels.

**Table 1 polymers-14-01768-t001:** Manufacturing parameters of laboratory-fabricated fiberboard panels bonded with adhesive systems composed of HL and PF resin.

Panel Type	Total Binders Content, %	PF Resin Content in the Adhesive System, %	Technical Hydrolysis Lignin Content in the Adhesive System, %	PF Resin Content Relative to Dry Fibers, %	Technical Hydrolysis Lignin Content Relative to Dry Fibers, %
A	10	10	90	1.0	9.0
B	10	20	80	2.0	8.0
C	10	30	70	3.0	7.0
D	12	10	90	1.2	10.8
E	12	20	80	2.4	9.6
F	12	30	70	3.6	8.4

**Table 2 polymers-14-01768-t002:** Density of fiberboard panels bonded with HL and PF resin.

Panel Type	Average/Mean/Value, kg·m^−3^	Standard Deviation, kg·m^−3^	Standard Error, kg·m^−3^	Coefficient of Variation, %	Probability, %
A	856	34	4	12	1.42
B	852	43	5	15	1.77
C	862	26	3	9	1.08
D	842	24	3	8	0.99
E	860	24	3	8	0.99
F	850	19	2	7	0.78

**Table 3 polymers-14-01768-t003:** ANOVA for the density of fiberboard panels bonded with HL and PF resin.

Source of Variation	SS	df	MS	F	*p*-Value	F_crit_
Total binder content	247.52	1	247.52	0.356	0.553	4.06
PF resin content in the adhesive system	393.16	2	196.58	0.282	0.755	3.21
Error	30,617.29	44	695.84			
Total	31,257.97	47				

**Table 4 polymers-14-01768-t004:** Regression models for the effect of total binder content and PF resin content in the adhesive system on the properties of the fiberboard panels produced in this work.

	Property	Modulus of Elasticity (MOE)	Bending Strength (MOR)	Internal Bond (IB) Strength	Thickness Swelling (24 h)	Water Absorption (24 h)
Regression Coefficient						
*B* _0_		−6920.667	−51.867	−2.260	60.538	111.152
*B* _1_		917.333	7.438	0.285	−2.493	−3.613
*B* _2_		230.450	0.395	0.018	−0.410	−0.530
*B* _12_		−17.550	–	−0.001	–	–
F_cal_		147.89	5882.68	974.77	37.892	29.039
F_(0.05,3,2)_		19.164	19.164	19.164	19.164	19.164

## Data Availability

The data presented in this study are available on request from the corresponding authors.

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
