# Peer review of "Effect of the Adhesive System on the Properties of Fiberboard Panels Bonded with Hydrolysis Lignin and Phenol-Formaldehyde Resin"

_polymers, 2022, doi:10.3390/polym14091768_

Round 1

Reviewer 1 Report

Dear authors,

The work presented proposed a greener alternative to typical commercial adhesives for fiberboards, however the results presented are quite limited to practical purposes (no chemical characterization or optical study via microscopy). In addition to that, there are some points that need improvement from the part of the authors:

-The abstract is rather long, try to modify it by being more direct and focus on the relevant results and conclusions.

-In the introduction section, Authors must highlight more clearly what is the innovation or main advantage of their work compared to other previous work from the literature. This point was not clear from the introduction section, since there are already various works dealing with this topic.

-Page 2, line 82: his sentence is not completely true. Even in the reference it does not say this categorically. The fact is that generally, a lower molecular weight is an advantage owing to a higher related reactivity (OH groups are more accesible).

-Page 3, lines 120-121: This part should be better included in the results and discussion section and further described and explained.

-Page 3, line132: On what base did the authors select these two percentages. Why not a higher or lower percentages?

-Page 5, lines 186-187: Did the authors performed the ANOVA analysis to confirm this?

-The results of MOE and MOR did not show any significant  beneficial effect of HL or considerable findings. In fact, as it was expected, when the amount of PF resins was increased in detriment of HL the properties were improved.

-Page 10, lines 327-328: Which effect is predominant for the improvement of the properties, the fact that total binder content was increased or the fact that the main component of binder was HL?
If the binder content is increased and the main componet was PF resin the improvement of the properties would be as good as with HL, better or worse?

-In general during the results and discussion section the parameters analyzed were interrelated so the tendencies observed were the same for all of them.

-Along the paper the advantage of using HL is not so evident. It seemed that the effect of a higher amount of binder in general and specially the addition of a higher amount of PF resins had a more predominant effect.

To sum up, I would appreciate that the author would be able to enahnce their study about the fiberboards using green adhesive by adding some strcutural/chemical analysis and optical ones (microscopy related ones) to see integration of components and actual influence of HL use as adhesive. Should these points be address I think this paper would be able to be considered for publication in the journal.

Author Response

Dear Reviewer,

Thank you for your time and efforts. We greatly appreciate your thoughtful comments that helped us improve the manuscript. Please find below our responses to your comments.

Point 1: "The work presented proposed a greener alternative to typical commercial adhesives for fiberboards, however the results presented are quite limited to practical purposes (no chemical characterization or optical study via microscopy)".

Answer:  Dear Reviewer, we fully agree with your observations. However, as one study cannot cover all issues on a given problem, and in view of the relevance of these issues and already significant volume of the paper, we envisage conducting a follow-up study with accompanying chemical and optical analyzes, including DSC heat of reaction thermograms of adhesive systems, TGA and DTG analysis etc.

Point 2: "The abstract is rather long, try to modify it by being more direct and focus on the relevant results and conclusions. "

Answer: Dear Reviewer, thank you very much for pointing out this issue. In the revised version of the manuscript the abstract is modified, according to your comments. The main novelty of the research is also presented: "This study aimed to propose an alternative technological solution for manufacturing fiberboard panels using a modified hot-pressing regime and hydrolysis lignin as the main binder. The main novelty of the research is the optimized adhesive system composed of unmodified hydrolysis lignin and reduced phenol-formaldehyde (PF) resin content. ".

Point 3: "In the introduction section, Authors must highlight more clearly what is the innovation or main advantage of their work compared to other previous work from the literature. This point was not clear from the introduction section, since there are already various works dealing with this topic."

Answer: Dear Reviewer, thank you for drawing our attention to the need for a more straightforward presentation of the study's novelty. Although we used the classical method for mixing PF resin with lignin, the advantage is the absence of modification and hence cost reduction. In the present study, we used a modified regime of hot-pressing - low with subsequent high pressure and cooling at high pressure. The modification of the hot-pressing regime aims for maximal utilization of the adhesive abilities of the lignin, in contrast to the classic hot-pressing cycles, which are developed for formaldehyde synthetic resins. This new approach is currently possible with a view to the widely introduction into the industry of continuous presses, which have autonomous heating of the individual sections. The main novelty is a derivation of the optimal composition of the adhesive system with maximum HL content when using a modified hot-pressing regime. That clarification is now given in the Introduction (lines 106-110).

Point 4: "Page 2, line 82: his sentence is not completely true. Even in the reference it does not say this categorically. The fact is that generally, a lower molecular weight is an advantage owing to a higher related reactivity (OH groups are more accesible)."

Answer: We fully agree with that and this sentence (statement) is now removed.

Point 5: "Page 3, lines 120-121: This part should be better included in the results and discussion section and further described and explained."

Answer: We completely agree with that. This part is now in the Results and discussion part, and the analysis to it has been expanded accordingly.

Point 6: "Page 3, line132: On what base did the authors select these two percentages. Why not a higher or lower percentages?"

Answer: Dear Reviewer, thank you very much for pointing out this shortcoming. "Due to the use of a bio-based binder, i.e. HL, the total binder content was higher compared to the conventional adhesive systems, composed only from thermosetting formaldehyde-based resins [64]. Thus, the total binder content used in this work was 10% and 12%. The higher content of binders when an adhesive system with the participation of lignin is used leads to a significant deterioration in the appearance of the panels, the need to extend the press factor and a very slight improvement in the properties of fiberboards [65, 66]". This part is now added to the manuscript. We also provided a more in-depth justification for the hot-pressing regime (lines 132-138).

Point 7: "Page 5, lines 186-187: Did the authors performed the ANOVA analysis to confirm this?"

Answer: Dear Reviewer, thank you very much for pointing out this omission. Yes, the ANOVA analysis confirmed that, and the result of this analysis is now included in the text (Table 3).

Point 8: "The results of MOE and MOR did not show any significant  beneficial effect of HL or considerable findings. In fact, as it was expected, when the amount of PF resins was increased in detriment of HL the properties were improved."

Answer: At the selected levels of variation of the total binder content and the content of PF resin in the adhesive system, the increase in the content of PF resin leads to an improvement in the MOE and MOR of the panels. However, it should be noted that in all experimental series, the HL is the main binder, and that is, a waste bio-based material is utilized. The improvement in the properties of fiberboards with increasing PF resin content should be due to better retention of lignin in the pulp and the interaction of lignin with PF resin. Which of the two causes dominates should be a subject of subsequent studies.

That clarification is also given in the revised version of the manuscript (lines 314-321).

Point 9: "Page 10, lines 327-328: Which effect is predominant for the improvement of the properties, the fact that total binder content was increased or the fact that the main component of binder was HL?
If the binder content is increased and the main component was PF resin the improvement of the properties would be as good as with HL, better or worse?"

Answer: Dear Reviewer, thank you very much for pointing us to the need of further clarification of that issue.

The predominant effect for improving the properties of fiberboard panels is the increase of total binder content, which increases both the content of HL and PF resin. However, the results show that the ratio of HL to PF resin is also significant. If the PF resin is the main binder, then in the used hot-pressing regime, the resin polymerization will occur at low pressure, i.e. before the final compression of the material has occurred. That will lead to fabricating panels with deteriorated physical and mechanical properties.

That clarification is now given in the revised version of the manuscript (lines 361-366).

Point 10: "In general during the results and discussion section the parameters analyzed were interrelated, so the tendencies observed were the same for all of them."

Answer: Yes, of course, the properties of the panels are interrelated. When the MOR improves, the MOE also improves, and when the water absorption decreases, the thickness swelling also reduces. However, a detailed analysis of all properties is necessary given the standardized requirements for them and, accordingly, to establish the values of the factors in which these requirements are met.

Point 11: "Along the paper the advantage of using HL is not so evident. It seemed that the effect of a higher amount of binder in general and specially the addition of a higher amount of PF resins had a more predominant effect."

Answer: The main advantage of using HL is the utilization of one bio-based residual product. We hope that with the corrections made so far in the manuscript, this advantage is more clearly outlined.

Point 12: "To sum up, I would appreciate that the author would be able to enahnce their study about the fiberboards using green adhesive by adding some strcutural/chemical analysis and optical ones (microscopy related ones) to see integration of components and actual influence of HL use as adhesive. Should these points be address I think this paper would be able to be considered for publication in the journal."

Answer: Dear Reviewer, we would like to thank you once again for pointing out shortcomings in our manuscript. We believe that most of these shortcomings have been rectified or addressed correctly, which has led to a significant improvement in the quality of our work.

Thank you very much for your time and consideration!

Reviewer 2 Report

After review the manuscript, it is a interesting work, and the use of biomaterials in areas such as adhesives can be important, the work also has a well structured form, just some observations/corrections details that need to be corrected, following they are detailed:

-in experimental section, please indicate how many replies for each sample were analyzed, also explain how was established the time for pressing process?

-In lines 232 and 275 indicate that grey zone in figure represents the limitation, but it is not indicated the figure number.

-In lines 237-240 make a comparison with Yotov work's, but which was the lignin content evaluated?

-It would be interesting to show a scheme of how is the interaction between HL and PF in fiberboard that causes the improvement in mechanical behavior, for a better understanding, I mean discuss about some reactions of physical interaction between fibers and lignin.

Author Response

Dear Reviewer,

Thank you for your kind words on our paper. We greatly appreciate your thoughtful comments that helped us improve the manuscript, and we trust that all your comments have been addressed accordingly in the revised version of the manuscript. Please find below our responses to your comments.

Point 1: in experimental section, please indicate how many replies for each sample were analyzed, also explain how was established the time for pressing process?”.

Answer:  Dear Reviewer, thank you very much for pointing out these shortcomings. Eight test samples were used to determine each property, and this clarification is now included in the manuscript.

The hot-pressing regime was chosen due to a previous study optimizing the pressing time using HL and PF resin [55]. In that study, an increase in the press factor for the second stage above 30 s.mm-1 is unjustified. Preliminary studies indicate that when an adhesive system from HL and PF resin is used, a deterioration in the waterproof properties of the panels is observed at hot-pressing temperatures below 200°. That is confirmed by other similar studies [65, 66]. The optimal parameters of the hot-pressing pressure have also been established experimentally. During the first stage, at a pressure above 1.0 MPa, it is difficult to separate the gas mixture from the panels. At a pressure below 4.0 MPa, it is difficult to compress the fibers to the final thickness and density of the panels. This clarification is now also added to the text.

Point 2: “In lines 232 and 275 indicate that grey zone in figure represents the limitation, but it is not indicated the figure number.”

Answer: Thank you very much for pointing out the omissions. The numbers of the figures are now indicated.

Point 3: “In lines 237-240 make a comparison with Yotov work's, but which was the lignin content evaluated?”

Answer:  The determined MOE values of laboratory-fabricated fiberboard panels with a hydrolysis lignin content of more than 10% in the adhesive system are in accordance with the results obtained by Yotov et al. This clarification is now added in the manuscript.

Point 4: “It would be interesting to show a scheme of how is the interaction between HL and PF in fiberboard that causes the improvement in mechanical behavior, for a better understanding, I mean discuss about some reactions of physical interaction between fibers and lignin.”

Answer: The PF resin and lignin structures are now given in the manuscript.

A discussion on the topic has also been added. As the paper is already in a significant volume and not all problems on a given topic can be presented in one study, no more in-depth discussion was given on the interaction of HL with phenol and formaldehyde and the interaction of the adhesive system with wood fibers. We are planning a follow-up study to make a comparative characterization of the different resins (UF resin, MF resin and PF resin) as auxiliary binders, and there these issues will be discussed in more detail.

Thank you very much for your time and consideration!

Reviewer 3 Report

The manuscript is focused on investigation of the effect of the total binder content and the optimal ratio between hydrolysis lignin and phenol-formaldehyde resin in the production of fiberboards using a modified hot-pressing regime. Overall, the manuscript is well-written, structured and informative, but still needs some minor improvements before acceptance for publication in the Polymers Journal. Please, see below my comments on your work:

The abstract (lines 12 to 29) and the keywords (lines 31-32) correspond to the specified aims and objectives of the manuscript. The abstract is concise, specific, and outlines the aim and main results of the study.

In the keywords, I recommend to add also “hot-pressing”.

The Introduction is very well prepared and provides relevant information on the research topic.

Materials and Methods: Overall, the Materials and Methods section is very well presented and provides relevant information about the materials used and methods applied in the experimental work. However, additional explanation is needed about the selected (modified) hot-pressing regime. Please explain the choice of hot-pressing parameters (temperature, specific pressure, and press duration).

In general, the Results section is properly developed and discussed with relevant research works in the field. Here I would recommend to provide data on the significance of statistical differences between the density values of the individual panels produced.

The Conclusion part reflects the main findings of the research work.

The references cited are appropriate and correspond to the research topic.

Best regards!

Author Response

Dear Reviewer,

Thank you for your kind words on our paper. We greatly appreciate your thoughtful comments that helped us improve the manuscript, and we trust that all your comments have been addressed accordingly in the revised version of the manuscript. Please find below our responses to your comments.

Point 1: The abstract (lines 12 to 29) and the keywords (lines 31-32) correspond to the specified aims and objectives of the manuscript. The abstract is concise, specific and outlines the aim and main results of the study. In the keywords, I recommend to add also “hot-pressing”.

Answer: Dear Reviewer, thank you again for the kind words on our paper. These comments of yours encourage us to continue our research in this area. Thank you for pointing us to this omission. The “hot-pressing” is now added to the keywords.

Point 2:The Introduction is very well prepared and provides relevant information on the research topic.

Answer: Thank you very much!

Point 3:Materials and Methods: Overall, the Materials and Methods section is very well presented and provides relevant information about the materials used and methods applied in the experimental work. However, additional explanation is needed about the selected (modified) hot-pressing regime. Please explain the choice of hot-pressing parameters (temperature, specific pressure, and press duration).

Answer: Dear Reviewer, thank you for pointing out these shortcomings in the Materials and Methods part.

The hot-pressing regime was chosen due to a previous study aimed at optimizing the pressing time using HL and PF resin [55]. In that study, an increase in the press factor for the second stage above 30 s.mm-1 is unjustified. Preliminary studies indicate that when an adhesive system from HL and PF resin is used, a deterioration in the waterproof properties of the panels is observed at hot-pressing temperatures below 200°. That is confirmed by other similar studies [65, 66]. The optimal parameters of the hot-pressing pressure have also been established experimentally. During the first stage, at a pressure above 1.0 MPa, it is difficult to separate the gas mixture from the panels. At a pressure below 4.0 MPa, it is difficult to compress the fibers to the final thickness and density of the panels.

The corresponding text is now included in the manuscript.

Point 4:In general, the Results section is properly developed and discussed with relevant research works in the field. Here I would recommend to provide data on the significance of statistical differences between the density values of the individual panels produced”.

Answer: Thank you very much for pointing out this omission. The Data from ANOVA for the density of fiberboard panels bonded with HL and PF resin are now included in the manuscript.

Point 5: “The Conclusion part reflects the main findings of the research work.”

Answer: Thank you very much!

Point 6: “The references cited are appropriate and correspond to the research topic.”.

Answer: Thank you very much!

Thank you very much for your time and consideration!

Round 2

Reviewer 1 Report

Dear authors,

Thank you for considering most of my commentaries and suggestions to improve your manuscript. I feel that after those correction the article would be able for publication. Nevertheless, I would have liked to find further improvements especially concerning the characterization of the panels (chemical, structural, morphology...). This point really would have given the work a higher scientific soundness and significance. Still the beneficial effect of HL is not clearly proven. In fact the results only showed that the formulations work for the industrial standards. I hope you could pursue this path to better study these missing parameters in the future.

Reviewer 2 Report

After review the corrected version the authors take in account the comments/observation and the manuscript shows a significante improve so I recommend Accept in present form.

Reviewer 3 Report

The manuscript has been thoroughly revised based on my previous comments and recommendations. I think it can be accepted in its current form.